# Establishing a Smartphone Ambulatory ECG Service for Patients Presenting to the Emergency Department with Pre-Syncope and Palpitations

**DOI:** 10.3390/medicina57020147

**Published:** 2021-02-06

**Authors:** Matthew J. Reed, Alexandra Muir, Julia Cullen, Ross Murphy, Valery Pollard, Goran Zangana, Sean Krupej, Sylvia Askham, Patricia Holdsworth, Lauren Davies

**Affiliations:** 1Emergency Medicine Research Group Edinburgh (EMERGE), Department of Emergency Medicine, Royal Infirmary of Edinburgh, 51 Little France Crescent, Edinburgh EH16 4SA, UK; alexandra.muir2@nhs.scot (A.M.); Julia.Cullen@nhslothian.scot.nhs.uk (J.C.); 2Acute Care Group, Usher Institute of Population Health Sciences and Informatics, College of Medicine and Veterinary Medicine, University of Edinburgh, Nine Edinburgh BioQuarter, 9 Little France Road, Edinburgh EH16 4UX, UK; 3Acute Medicine, NHS Lothian, Edinburgh EH16 4SA, UK; Ross.Murphy@nhslothian.scot.nhs.uk (R.M.); Valery.Pollard@nhslothian.scot.nhs.uk (V.P.); Goran.Zangana@nhslothian.scot.nhs.uk (G.Z.); Sean.Krupej@nhslothian.scot.nhs.uk (S.K.); Sylvia.Askham@nhslothian.scot.nhs.uk (S.A.); Patricia.Holdsworth@nhslothian.scot.nhs.uk (P.H.); Lauren.Davies@nhslothian.scot.nhs.uk (L.D.)

**Keywords:** emergency department, diagnosis, ECG monitoring, cardiac dysrhythmias

## Abstract

*Background and Objectives:* The Investigation of Palpitations in the ED (IPED) study showed that a smartphone-based event recorder increased the number of patients in whom an electrocardiogram (ECG) was captured during symptoms over five-fold to more than 55% at 90 days compared to standard care and concluded that this safe, non-invasive and easy-to-use device should be considered part of on-going care to all patients presenting acutely with unexplained palpitations or pre-syncope. This study reports the process of establishing a smartphone palpitation and pre-syncope ambulatory care Clinic (SPACC) service. *Materials and Methods:* A clinical standard operating procedure (SOP) was devised, and funding was secured through a business case for the purchase of 40 AliveCor devices in the first instance. The clinic was launched on 22 July 2019. *Results:* Between 22 July 2019 and 31 October 2019, 68 patients seen in the emergency departments (EDs) with palpitations or pre-syncope were referred to SPACC. Of those, 30 were male and 38 were female, and the mean age was 45.8 years old (SD 15.1) with a range from 18 years old to 80 years old. A total of 50 (74%) patients underwent full investigation. On the first assessment, seven (10%) patients were deemed to have non-cardiac palpitations and were not fitted with the device. All patients who underwent full investigation achieved symptomatic rhythm correlation most with sinus rhythm, ventricular ectopics, or bigeminy. A symptomatic cardiac dysrhythmia was detected in six (8.8%) patients. Three patients had supraventricular tachycardia (4%), two had atrial fibrillation (3%), and one had atrial flutter (2%). Qualitative feedback from the SPACC team suggested several areas where improvement to the clinic could be made. *Conclusion:* We believe a smartphone palpitation service based on ambulatory care is simple to implement and is effective at detecting cardiac dysrhythmia in ED palpitation patients.

## 1. Introduction

Patients with palpitations and pre-syncope commonly present to emergency departments (EDs), accounting for 300,000 ED presentations a year in the United Kingdom [1,2] and being one of the commonest presentations to general and family practice (16% of presentations) [3].

Diagnosing the underlying rhythm can be difficult and is commonly not possible during the ED visit. Diagnosis of the underlying heart rhythm requires an electrocardiogram (ECG) to be recorded while the patient is symptomatic. However, 12-lead ECG is of limited efficacy and conventional ambulatory monitoring such as Holter has a diagnostic yield of less than 20% mainly due to the infrequency of symptoms [4].

The AliveCor/Kardia mobile technology is a Food and Drug Administration (FDA) cleared, CE (Conformitè Europëenne) marked single-lead rhythm strip comparable to lead I of standard ECG machines and is the most clinically validated ambulatory ECG device available worldwide. When used alongside the Kardia app, the device provides instant analysis for normal sinus rhythm, atrial fibrillation, sinus bradycardia, and sinus tachycardia in around 30 s [5].

The Investigation of Palpitations in the ED (IPED) study [6] was a multi-centre, open-label, and randomised controlled trial. A total of 243 adults (≥16 years old) presenting to 10 United Kingdom (UK) hospital EDs were randomised over an 18-month period to either (a) an intervention group receiving standard care alongside a smartphone-based event recorder or (b) a control group receiving standard care alone. The primary endpoint was the detection of a symptomatic rhythm at 90 days.

The results showed that the AliveCor/Kardia smartphone-based event recorder increased the number of patients recording an ECG during symptoms (symptomatic rhythm) over five-fold at 90 days (69/124; 55.6%; 95% Confidence Interval (CI) 46.9–64.4% versus 11/116; 9.5%; 95% CI 4.2–14.8%; Relative Risk (RR) 5.9, 95% CI 3.3–10.5; *p* < 0.0001). The mean time to detecting a symptomatic rhythm in the intervention group was 9.5 days (SD 16.1, range 0–83) compared to 42.9 days (SD 16.0, range 12–66; *p* < 0.0001) in the control group. Sinus rhythm, sinus tachycardia, and ectopic beats were the commonest symptomatic rhythms detected. A symptomatic cardiac dysrhythmia was detected at 90 days in 11 (*n* = 124; 8.9%; 95% CI 3.9–13.9%) participants in the intervention group compared to one (*n* = 116; 0.9%; 95% CI 0.0–2.5%) in the control group (RR 10.3, 95% CI 1.3–78.5; *p* = 0.006).

The IPED study concluded that the use of a smartphone-based event recorder such as AliveCor/Kardia is safe, non-invasive, and easy to use and should be considered for all patients presenting acutely to ED with unexplained palpitations or pre-syncope. This study reports the subsequent establishment of a smartphone palpitation and pre-syncope ambulatory care clinic (SPACC).

## 2. Materials and Methods

A clinical standard operating procedure (SOP) was devised, and funding was secured through a business case for the purchase of 40 AliveCor devices in the first instance which are able to be cleaned and reused multiple times. From 22 July 2019, all patients aged 16 years or older presenting to the ED or Acute Medicine Unit (AMU) of the Royal Infirmary of Edinburgh (RIE) with palpitations or pre-syncope, whose ECG was normal, who had a compatible Apple/android phone, tablet, or watch, and in whom an underlying cardiac dysrhythmia was possible, were offered an appointment at the SPACC, which was based in an ambulatory care clinic setting beside the ED. Ambulatory care is a service which offers same or next-day, hospital-based emergency and acute care, meaning that patients are assessed, diagnosed, treated, and are able to go home without being admitted into a hospital bed overnight wherever possible.

Exclusion criteria included the patient being non-ambulant, requiring hospital admission, having a prior diagnostic ECG, having multiple frequent episodes or recent acute myocardial infarction (AMI), severe heart failure, or unstable angina, having associated chest pain or syncope, being unwilling or unable to use the AliveCor Heart Monitor and ECG App, having a cardiac pacemaker or other implanted electronic device, or having a likely non-cardiac cause for their palpitations (e.g., anxiety, sepsis).

The patient’s phone, tablet, or watch was checked for compatibility, and they were asked to bring their smartphone, tablet, or watch and app store password to the ambulatory appointment (and later were asked to download the Kardia app prior to coming to the clinic but not to set it up, which was done in the clinic). Routine blood tests including thyroid function tests, full (complete) blood count, urea and electrolytes, and magnesium levels were taken, and the patient was then discharged with a patient advice leaflet to be seen in the SPACC on the next available day. Initially, only ED and AMU referrals were taken. The IPED study [6,7] showed that 93% of participants recording a symptomatic rhythm during the 90 days, did so in the first 28 days. It was therefore decided that patients would be reviewed at four weeks to enable efficient device usage and timely treatment. Patients without a symptomatic rhythm at four weeks could be re-reviewed if necessary.

Other components of the SPACC SOP were a list of compatible devices for the ED/AMU clinician to refer to, a patient symptom diary, a patient instruction manual, a clinic checklist, and advice for the clinic clinician on how to incorporate a patient ECG into the RIE electronic patient record (EPR). We also sought approval from our hospital data controller (termed Caldicott Guardian) who suggested using anonymised patient information, standardised for all patients, in the Kardia application (i.e., first name ‘ambulatory’, last name ‘care’, date of birth ‘01/01/1980′). The local ethics service deemed the study to be a service evaluation and therefore formal ethical approval was not required. The study was registered on the RIE ED Quality Improvement Project (QIP) database. A data template was created using REDCap, a secure electronic database (http://www.project-redcap.org (accessed on 3 January 2021)) for anonymised data entry, [8,9] which was funded by a grant from the Royal College of Emergency Medicine (RCEM).

## 3. Results

Between 22 July 2019 and 31 October 2019, 68 patients were seen in the ED with palpitations or pre-syncope and were referred to SPACC. Table 1 details their baseline characteristics. Of those, 30 were male and 38 were female, and the mean age was 45.8 years old (SD 15.1) with a range from 18 years old to 80 years old. Figure 1 details the flow of patients through the SPACC. A total of 50 (74%) patients underwent full investigation. On the first assessment, seven (10%) patients were deemed to have non-cardiac palpitations and were not fitted with the device. A symptomatic cardiac dysrhythmia was detected in six (8.8%) patients. Three patients had supraventricular tachycardia (SVT; 4%), two had atrial fibrillation (3%), and one had atrial flutter (2%). All other patients undergoing investigation had a non-cardiac symptomatic rhythm detected during their SPACC investigation period with sinus rhythm, ventricular ectopics, and bigeminy being detected.

### Difficulties Addressed and Improvements Made

Clinic referral criteria—qualitative feedback from the SPACC team suggested several areas where improvement to the service could be made. Firstly, it was noted that 10% of patients referred to the clinic were deemed on the first assessment to have non-cardiac palpitations and were not fitted with the device. A pre-planned sub-study of the IPED study [10] asked the treating ED clinician to rate the likelihood of underlying cardiac dysrhythmia ranging from 1 (least likely) to 10 (most likely). An ED clinician likelihood rating of 5 or more had 92% sensitivity and 59% specificity for predicting cardiac dysrhythmia. This sub-study concluded that ED clinicians are able to predict the likelihood of cardiac dysrhythmia in patients presenting to the ED with palpitation or pre-syncope with reasonable accuracy. It was therefore decided to review the SPACC referral criteria to ensure the clinic slots were prioritised for patients ‘thought to be at risk of cardiac dysrhythmia’.

Patients expectations—some patients were coming to the SPACC with an expectation that they were going to be fitted with the AliveCor device. In order to manage patients’ expectations when the clinic staff might feel that the risk of cardiac dysrhythmia was not high enough to warrant AliveCor device fitting, the referral pathway was revised to ensure that patients were counselled in the ED that they were coming to the SPACC for assessment for AliveCor device fitting.

Embedding of electronic ECGs into the electronic patient record—it was felt that better embedding of electronic ECGs into the EPR was required. A process and protocol for uploading ECGs into EPRs were therefore developed.

ECG interpretation—occasionally when the recorded ECG included noise or artefact, less experienced clinic staff had difficulty interpreting the ECG and would be more likely to order additional investigations or further AliveCor wear time, whereas more senior clinicians were comfortable interpreting these recorded ECGs as normal sinus rhythm. It was therefore encouraged for staff to seek a second opinion from the ED or on-call medical consultant if required, and an ECG diagnostic algorithm was also developed.

Downloading the Kardia app—due to poor Wi-Fi and phone reception in the SPACC location, it was decided to alter the referral pathway to ensure patients were asked to download the Kardia app prior to clinic attendance to increase clinic efficiency.

Length of time between clinic appointments—the median time patients had the device for was 28 days (Q1, Q3 15.25–30 days). Although the IPED study experience deemed that this was the optimum time, the SPACC staff felt occasionally that this was too long for patients who had recorded one of the more symptomatic rhythms within the first week or two of having the device. In order to optimise AliveCor device turn over and to counsel patients who had already recorded a symptomatic rhythm it was decided to bring patients back earlier at two weeks and if no symptomatic rhythm had been recorded, to allow them to continue using the device until this had occurred, which for one patient was 76 days.

Other changes to the clinic protocol that were made included less emphasis on patient diaries which were often poorly completed, better logging of devices, and a routine battery change for every device every 12 months.

## 4. Discussion

This is the first report anywhere of an ambulatory smartphone palpitation and pre-syncope service for emergency and acute medical patients presenting with pre-syncope and palpitations. The service allows patients who present to either the ED or the AMU of the RIE with pre-syncope and palpitations to be referred to a next-day assessment clinic for consideration of AliveCor/Kardia device fitting [11].

Our preliminary three-month clinic data show that the detection of symptomatic cardiac dysrhythmia in 8.8% of patients is comparable to the 8.9% of patients who had a symptomatic cardiac dysrhythmia detected in the IPED study [6] and show that a research protocol and research finding can be successfully extrapolated and implemented in a pragmatic clinical setting.

We plan to continue to assess our service and to evaluate further the effect of the changes that were made at the three-month point. Further work could include scaling up the clinic in order to allow general and family practitioners to refer patients for investigation and also to integrate the service further with our hospital’s cardiology service. We have also given support to other health boards across the United Kingdom to help them establish a similar service and contributed to National Institute for Health and Care Excellence (NICE) evaluations both in England and Wales of this service model [5,12].

At a time of rapid expansion of health-related applications and smartphone use, this model of remote monitoring will allow more personalised care and a reduction in the need for in-hospital care. Bringing back patients to an ambulatory care setting allows assessment and patient education to take place in a less chaotic environment than in the ED. The rapid detection of non-life-threatening dysrhythmias such as atrial fibrillation (AF) and supraventricular tachycardia (SVT) from the ED also potentially reduces ED attendances, reduces the need for unnecessary cardiology referrals and investigations, allows earlier prophylactic treatment of those at risk of AF associated cerebrovascular accident (CVA), and allows earlier treatment of those with symptomatic SVT.

Our service model is generalisable to a wide range of healthcare systems and the emergency and acute setting and could equally be applied to general and family practitioner settings for less acute patients. The existence of an AliveCor postal service also allows hospitals to send out devices which is important during the recent coronavirus disease 2019 (COVID-19) pandemic which is likely to be an ongoing concern in healthcare provision for at least the next 12 months. This remote option reduces the need for repeat face to face out-patient clinic attendances and may potentially increase the efficiency of diagnosis and definitive treatment.

## 5. Conclusions

We believe a smartphone ambulatory ECG palpitation service is simple to implement and is effective at detecting cardiac dysrhythmia in emergency and acute palpitation and pre-syncope patients.

## Figures and Tables

**Figure 1 medicina-57-00147-f001:**
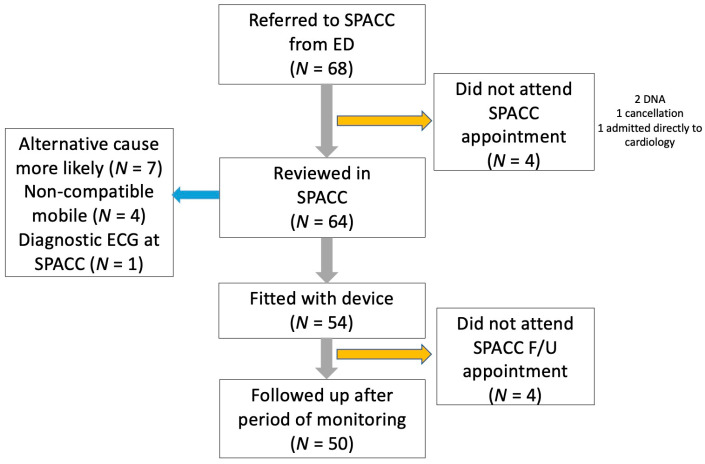
Flowchart of patients through the smartphone palpitation and pre-syncope ambulatory care clinic (SPACC); ED = Emergency Department; DNA = Did not attend; F/U = Follow up.

**Table 1 medicina-57-00147-t001:** Baseline characteristics of patients referred to smartphone palpitation and pre-syncope ambulatory care clinic (SPACC) (*n* = 68).

Characteristic	*n* (%) Unless Specified Otherwise
Gender	Male 30 (44.1%), Female 38 (55.9%)
Age	45.8 (SD 15.1)
**Predominant presenting symptom**	
Fluttering or racing	38 (55.9%)
Chest pain or pressure	9 (13.2%)
Skipped /missed beat(s)	5 (7.4%)
Pounding	4 (5.9%)
Fainted	3 (4.4%)
Lightheaded	3 (4.4%)
Arm or neck pain/tingling	2 (2.9%)
Dizziness	2 (2.9%)
Irregular beating	2 (2.9%)
**Presenting symptom duration**	
1 min or less	5 (7.4%
10 min or less	21 (30.9%)
1 h or less	26 (38.2%)
More than 1 h	16 (23.5%)

## Data Availability

The data presented in this study are available on request from the corresponding author.

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
