# Peer review of "Establishing a Smartphone Ambulatory ECG Service for Patients Presenting to the Emergency Department with Pre-Syncope and Palpitations"

_medicina, 2021, doi:10.3390/medicina57020147_

Round 1
Reviewer 1 Report
Establishing a smertphone ambulatory ECG service for patients presenting to the emergency department with presyncope and palpitations / reviewer comments
This paper, written by Reed et al, is based on published IPED trail, in which they have shown the advantage of using a smartphone-based event recorder in addition to standard evaluation of patients with palpitations and presyncope. In the current paper, they try to show the process of establishing a Smartphone-recorder service in real practice.
The majority of the current trial basically repeats the experience of the IPED trial showing again almost identical percent of patients in whom a symptomatic arrhythmia could be detected. Only minority of the paper is dedicated to the issue of establishing such service.
In this minor part of clinic establishment they report the following issues:
- Using an ED clinician to exclude patients in whom non-cardiac palpitations are expected, showing their clinical judgment to be pretty good.
- Adjusting patients' expectations for the service, so they are not promised to be fitted with the device but rather assessment for device fitting.
- Having patients download the device application to their Smartphone at home, in order to increase clinic efficiency.
- Dealing with issues of ECG interpretation by having a second opinion from a senior staff member.
On the whole, the paper adds little to the issue and most of the above points are quite straightforward. Furthermore, it is not clear what is the advantage of this device over the commonly used external long-term ECG monitors which are used world-wide these days.
Thus, I do not see a major novel issue in this manuscript
Author Response
Thank you for reviewing our manuscript. In response we would respectively point out that the submitted paper is not reporting a trial but in fact an implementation study into real world practice of an intervention shown to be effective in the IPED study. This is the first reported implementation of the AliveCor event recorder device into Emergency practice. As I am sure the author is aware, event recorders are much more suited to investigating palpitations as the patient is conscious throughout the episode, as opposed to syncope which requires external long-term ECG monitors as the author suggests. External long-term ECG monitors are more expensive, cumbersome to wear continuously and not necessary in order to investigate the symptom of palpitations as we have successfully shown in our study.
There are no other specific points raised by the reviewer to address.
Reviewer 2 Report
In the manuscript “Establishing a smartphone ambulatory ECG service for patients presenting to the Emergency Department with pre-syncope and palpitations”, the authors aimed to investigate the applicability and reliability of ECG monitoring using "AliveCor" devices in patients with palpitations or syncope. The topic of the manuscript is certainly interesting and captures one of the questions of the moment remote monitoring during COVID-19 pandemic. The authors state that smartphone palpitation service based in ambulatory care is simple to implement and is effective at detecting cardiac dysrhythmia.
However, there are some points that need further clarification:
- Please remove citation [1] from abstract
- Please, you should explain each of your abbreviations the first time it appears in the main text(i.e. IPED, COVID, ect.).
- The authors state in the methods section that 40 devices were purchased, but then 68 patients are enrolled. Please specify this point.
- Please double-check the percentages (e.g., 7/68= 10%, not 11%).
- Please double-check the percentages (e.g., 7/68= 10%, not 11%). Please move some of the considerations reported in the results to the methods section and better discuss the impact of remote monitoring might have in clinical practice.
- Please provide a table with demographic characteristics of enrolled patients and their medications.
Author Response
- Please remove citation [1] from abstract
This has been done
- Please, you should explain each of your abbreviations the first time it appears in the main text(i.e. IPED, COVID, ect.).
This has been done
- The authors state in the methods section that 40 devices were purchased, but then 68 patients are enrolled. Please specify this point.
Devices are able to be reused multiple times. This has been clarified in the text (see line 81).
- Please double-check the percentages (e.g., 7/68= 10%, not 11%).
Thank you. This has been corrected in the text (see lines 25 and 127)
- Please double-check the percentages (e.g., 7/68= 10%, not 11%).
Thank you. This has been corrected in the text (see lines 25 and 127)
- Please move some of the considerations reported in the results to the methods section and better discuss the impact of remote monitoring might have in clinical practice.
We have looked at this However wonder whether these considerations are in fact all part of the qualitative part of the study and therefore are correctly placed in the results section. It would be difficult to move some sections of the ‘Difficulties addressed and improvements made’ section to the methods without moving the whole lot?
We have further discussed the impact of remote monitoring on clinical practice in the discussion section (lines 204-212): ‘At a time of rapid expansion of health-related applications and smartphone use, this model of remote monitoring will allow more personalized care, and a reduction in need for in hospital care. Bringing back patients to an ambulatory care setting allows assessment and patient education to take place in a less chaotic environment than in the ED. The rapid detection of non-life threatening dysrhythmias such as AF and SVT from the ED also potentially reduces ED attendances, reduces the need for unnecessary cardiology referrals and investigations, allows earlier prophylactic treatment of those at risk of AF associated Cerebrovascular Accident (CVA) and allows earlier treatment of those with symptomatic SVT.’
- Please provide a table with demographic characteristics of enrolled patients and their medications.
We have now provided a table with the demographic characteristics of the enrolled patients that we collected. We did not collect patients’ medications prior to clinic.
Round 2
Reviewer 1 Report
Understood
Author Response
There are no further comments to address. Thank you.
Reviewer 2 Report
In the revised version the paper is improved. Please provide characteristics at baseline for patients regarding risk factors and therapies.
Author Response
Reviewer 2: In the revised version the paper is improved. Please provide characteristics at baseline for patients regarding risk factors and therapies.
Authors reply: We provided a table with demographic characteristics at baseline of the enrolled patients in the first revision and have supplied age and gender information. Unfortunately we did not collect characteristics of the enrolled patients (e.g. medication or other risk factors) other than age and gender.